# Accuracy of Six Intraocular Lens Power Calculations in Eyes with Axial Lengths Greater than 28.0 mm

**DOI:** 10.3390/jcm11195947

**Published:** 2022-10-08

**Authors:** Majid Moshirfar, Kathryn M. Durnford, Jenna L. Jensen, Daniel P. Beesley, Telyn S. Peterson, Ines M. Darquea, Yasmyne C. Ronquillo, Phillip C. Hoopes

**Affiliations:** 1Hoopes Vision, HDR Research Center, Draper, UT 84020, USA; 2John A. Moran Eye Center, Department of Ophthalmology and Visual Sciences, Salt Lake City, UT 84132, USA; 3Utah Lions Eye Bank, Murray, UT 84107, USA; 4School of Medicine, University of Utah, Salt Lake City, UT 84132, USA; 5Brigham Young University, Provo, UT 84602, USA; 6College of Osteopathic Medicine, Rocky Vista University, Ivins, UT 80112, USA

**Keywords:** IOL accuracy, high myope, high axial length, Caucasian, Kane, Barrett, EVO, Hill-RBF, Holladay 1, SRK/T

## Abstract

The purpose of this study was to compare the accuracy of several intraocular (IOL) lens power calculation formulas in long eyes. This was a single-site retrospective consecutive case series that reviewed patients with axial lengths (AL) > 28.0 mm who underwent phacoemulsification. The Wang–Koch (WK) adjustment and Cooke-modified axial length (CMAL) adjustment were applied to Holladay 1 and SRK/T. The median absolute error (MedAE) and the percentage of eyes with prediction errors ±0.25 diopters (D), ±0.50 D, ±0.75 D, and ±1.00 D were used to analyze the formula’s accuracy. This study comprised a total of 35 eyes from 25 patients. The Kane formula had the lowest MedAE of all the formulas, but all were comparable except Holladay 1, which had a significantly lower prediction accuracy with either AL adjustment. The SRK/T formula with the CMAL adjustment had the highest accuracy in predicting the formula outcome within ±0.50 D. The newer formulas (BU-II, EVO, Hill-RBF version 3.0, and Kane) were all equally predictable in long eyes. The SRK/T formula with the CMAL adjustment was comparable to these newer formulas with better outcomes than the WK adjustment. The Holladay 1 with either AL adjustment had the lowest predictive accuracy.

## 1. Introduction 

The accurate calculation of intraocular lens (IOL) power is critical for providing optimal visual acuity results for cataract surgery patients. As axial lengths reach the extremes, the variation in outcomes increases significantly, demonstrating the need to carefully select the best formula [1,2,3]. The first-generation SRK I, second-generation SRK II, and Hoffer formulas have given way to the more modern third-generation Holladay 1, Hoffer Q, and SRK/T formulas, as well as the fourth-generation Haigis and Barrett Universal II (BU-II) formulas [3,4]. The SRK/T formula, in particular, has been shown to be particularly accurate in eyes with an AL ≥ 27.0 mm [5,6,7]. More recent formulas include the Hill–Radial Basis Function version 3.0, Emmetropia Verifying Optical (EVO), and Kane [7,8,9]. The resulting improvement in accuracy has increased the popularity of newer formulas among cataract surgeons [1,5,10].

Despite the increased accuracy of the newer third- and fourth-generation formulas, they tend to underestimate IOL power for patients with longer eyes, causing postoperative hyperopia [11]. An AL modification method, referred to as the Wang–Koch (WK) adjustment, was published by Wang et al. in 2017, which increases the accuracy of older generation formulas in patients with high AL [11]. Fernández et al. demonstrated that the variation in prediction error (PE) with axial length was due to considering a single refractive index and not due to errors in the prediction of effective lens position (ELP), suggesting a variation in the fictitious refractive index to address this problem [12]. In 2019, the Cooke-modified axial length (CMAL) method was proposed, which sums the individual ocular segment lengths to predict a sum-of-segments AL that improved the predictive power of third- and fourth-generation formulas in long and short eyes [13]. The addition of these methods increased the prediction accuracy of newer generation formulas such as SRK/T and Holladay in highly myopic eyes [5,14].

The BU-II formula was first presented as a modified version of the original Barrett formula in 2010; it is considered one of the most accurate but remains unpublished [5,9]. Other unpublished formulas include the EVO (Tun Kuan Yeo, MD), Kane (Jack Kane, MD), and Hill-RBF version 3.0 [9,15]. The Kane and Hill-RBF version 3.0 formulas both incorporate artificial intelligence to predict IOL power [7]. The newly updated Hill-RBF version 3.0 increased the database for eyes of all sizes, including myopic patients, and added central corneal thickness (CCT), lens thickness (LT), white-to-white (WTW), and gender to the existing parameters [15].

To our knowledge, the vast majority of studies evaluating the accuracy of the newer formulas in highly myopic patients have taken place with Asian participants, where high myopia is more common [16,17]. As eyes in Asian populations also tend to have flatter corneas, it is possible that the use of these formulas in other racial groups with longer axial lengths could have differing results [18,19]. Given this paucity of data outside of Asian populations, there is a need for studies of these newer formulas in highly myopic patients of non-Asian descent. The purpose of this study was to compare the IOL accuracy of six IOL power formulas for eyes with extremely long axial lengths (≥28.0 mm) from a predominantly Caucasian population.

## 2. Materials and Methods

### 2.1. Subjects and Procedures

This study was a retrospective review of consecutive cataract patients having undergone uncomplicated phacoemulsification procedures at a single site from January 2013 to May 2021. The Lenstar^®^ LS 900 (Haag-Streit AG, Koeniz, Switzerland) reviewed 16,538 eyes and initially identified 71 records of patients with axial lengths > 28.0 mm who subsequently underwent simple phacoemulsification. Patients included had a manifest refraction performed at least one month postoperatively with a corrected distance visual acuity (CDVA) of 20/40 or better. Eyes with prior refractive surgery, intraoperative or postoperative cataract complications, a history of severe fundus pathology (e.g., myopic degeneration or macular hole), or a lack of postoperative refraction at one month or greater were excluded; this left 35 eyes of 25 patients for analysis (Appendix A). The data were de-identified prior to analysis.

Biometric measurements included AL, keratometry (K), anterior chamber depth (ACD, measured from the corneal epithelium to the lens), WTW, CCT, LT, and aqueous depth. Preoperative refraction, uncorrected distance visual acuity (UDVA), and CDVA were noted as well. All visual acuity measurements were converted to the equivalent logarithm of the minimum angle of resolution (LogMAR). Additionally, the patient’s age, gender, and past ocular and medical history were included in the data analysis. The patient’s self-reported ethnicity data were not recorded in the paper charts. Ethnicity was projected from the area’s demographics and recent census data.

### 2.2. Surgery and Intraocular Lenses

The standard phacoemulsification procedure was performed on all the patients by a single experienced surgeon using topical anesthesia. One of six different foldable acrylic lenses was inserted into the eye. The models of IOL lens choices included the AR40e (Sensar®, Johnson & Johnson Vision, Jacksonville, FL, USA), MA60MA (Alcon Laboratories, Fort Worth, TX, USA), MX60E (enVista®, Bausch + Lomb, Rochester, NY, USA), ZCB00 (TECNIS®, Johnson & Johnson Vision, Jacksonville, FL, USA), ZCT225 (TECNIS®, Johnson & Johnson Vision, Jacksonville, FL, USA), and ZXR00 (TECNIS Symfony®, Johnson & Johnson Vision, Jacksonville, FL, USA). Patients were instructed to use third- or fourth-generation fluoroquinolone antibiotic eye drops four times daily for one week. Patients were also started on a topical steroid medication four times daily and tapered weekly over one month. A topical NSAID eye drop was used twice daily for six weeks. Patients were then scheduled for one-day, one-week, and one-month follow-up appointments where the LogMAR, UDVA, and intraocular pressure (IOP) were checked. At the one-month postoperative appointment, manifest refraction was performed, and LogMAR CDVA was recorded. Any additional follow-up appointments were recorded within two years; IOP, CDVA, UDVA, and manifest refractions were recorded for any of these follow-up visits. The last manifest refraction charted was recorded as the patient’s postoperative spherical equivalent (SE) to analyze the prediction error.

### 2.3. Retrospective and Statistical Analysis

The six formulas evaluated were:(1)Barrett Universal II (available at https://calc.apacrs.org/barrett_universal2105/, hereafter referred to as BU-II, accessed 12 September 2021);(2)Emmetropia Verifying Optical (available at https://www.evoiolcalculator.com/, referred to as EVO, accessed 12 September 2021);(3)Hill–Radial Bias Function 3.0 Calculator (available at https://rbfcalculator.com/, hereafter referred to as Hill-RBF, accessed 12 September 2021);(4)Holladay 1 [20];(5)Kane (available at https://www.iolformula.com/, accessed 12 September 2021);(6)SRK/T [21].

Due to the small study sample and evaluation of long eyes, IOL constants for the implanted lenses were obtained from those listed in the User Group for Laser Interference Biometry database (available at http://ocusoft.de/ulib/c1.htm, referred to as ULIB, accessed 15 June 2021). To calculate the predicted SEs, the ULIB IOL constants for each lens and the patient’s biometric data were input into each formula. The published Holladay 1 and SRK/T formulas were exported directly into Excel (Microsoft Corporation, Redmond, WA). The AL was adjusted for the Holladay 1 and SRK/T formulas using the modified regression WK adjustment and the CMAL adjustment, resulting in two iterations of each formula. For the remaining four unpublished formulas, ULIB IOL constants and biometrics were input into the online calculators via Python (Python Software Foundation, Wilmington, DE) as recommended by Hoffer and Savini [22]. Refractive prediction errors (PE) were then calculated by subtracting the formula-predicted SE refractive error from the postoperative manifest refraction SE. Absolute prediction errors (AE) were calculated from the PE.

Mean prediction error (MPE) was used to assess for postoperative myopic or hyperopic surprises. As described by Hoffer and Savini, the mean absolute error (MAE) and median absolute error (MedAE) were calculated for each formula to assess the predictive accuracy [22]. The max AE was noted for each formula as well. The cases were analyzed by the percentage of eyes with a PE of ±0.25 diopters (D), ±0.50 D, ±0.75 D, and ±1.00 D for each formula. The postop refraction values were rounded to the closest step to account for the invariant refraction assumption.

Statistical analysis was performed using R version 4.0.2 (22 June 2020) statistical software. Continuous variables were reported with a mean and standard deviation (SD), and categorical variables were reported with a number and percentage. The normality of the data was assessed with the Shapiro–Wilk test. The distribution of the AE values did not follow normal Gaussian distribution, so nonparametric tests were used. The difference in refractive errors between formulas was assessed with the paired Wilcoxon signed-rank test. The percentages of PE within ±0.25 D, ±0.50 D, ±0.75 D, and ±1.00 D for each formula were compared with a Cochran’s Q test. McNemar’s test with Bonferroni adjustment was then used to identify any statistically significant difference identified in the Cochran’s Q. A *p*-value less than 0.05 was considered statistically significant.

## 3. Results

### 3.1. Population Demographics

The study comprised 35 eyes of 25 patients. The mean age of the study was 56.94 ± 9.56 years and 60% of the patients were female. The mean axial length was 28.71 ± 0.87 mm (range: 28.01 mm to 31.10 mm). The mean corneal power was 43.30 ± 1.61 D, with the majority of patients’ average keratometry (65.7%) falling between 42.0 D and 46.0 D. The average ACD was 3.66 ± 0.38 mm and the average LT of the patients was 4.25 ± 0.52 mm. Table 1 summarizes all the biometrics, refractive outcomes, and demographics of the study population.

The refractive measurements show a marked improvement in refractive error following phacoemulsification surgery. The mean CDVA before the procedure was 0.22 ± 0.17 (LogMAR) and improved to 0.01 ± 0.07 (LogMAR) after the procedure. After surgery, the mean SE approached emmetropia at –0.58 ± 0.79 compared to a mean preoperative SE that was highly myopic at –11.28 ± 4.29. The implanted IOLs had a mean power of 7.76 ± 3.06 D (range: –1.00 to +12.00 D) with only one (2.86%) implanted lens with a minus power diopter (Table 1).

### 3.2. Accuracy of the Six Formulas

Figure 1A shows the distribution and interquartile ranges of the PEs for each formula, and Table 2 shows the MPE of each formula to illustrate the tendency of each formula to lead to either myopic or hyperopic surprises. The four newer formulas tended to result in hyperopic surprises as compared with the older SRK/T and Holladay 1 formulas. The WK AL adjustment tended to lead to a myopic surprise for both SRK/T and Holladay 1. In contrast, the CMAL adjustment tended towards a hyperopic shift for Holladay 1. Interestingly, the median value of PE closest to zero was the SRK/T formula with CMAL adjustment, and this formula’s MPE is also closest to zero. Kane’s formula had the lowest MedAE (0.270), with the three newer formulas (BU-II, EVO, and Hill-RBF) following closely behind (Table 2). The WK adjustment, with either SRK/T or Holladay 1, had the highest MedAE values. The WK adjustment for SRK/T had the highest MedAE of all the formulas, but the CMAL adjustment with SRK/T was not far from the BU-II formula. The Wilcoxon signed-rank test revealed a statistically significant difference (*p* < 0.05) of the WK adjustment to either SRK/T and Holladay 1 compared to the CMAL adjustment of SRK/T, indicating that the WK adjustment, with either formula, had lower accuracy (Table 3). Figure 1B shows a boxplot of the AEs for each formula. The Holladay 1 with WK adjustment and SRK/T with WK adjustment had the widest range and highest MedAE compared with the CMAL adjustments or the four newer formulas.

Cochran’s Q test evaluated the percentage of eyes within ±0.25 D, ±0.50 D, ±0.75 D, and ±1.00 D. The percentages for each formula are stated in Table 2 and graphically represented in Figure 2. The only significant difference among the formulas was found between the percentage of eyes within ±0.50 D and ±0.75 D. Further testing identified that the statistically significant difference was between the SRK/T-CMAL formula compared to the SRK/T-WK, Holladay 1-CMAL, and Holladay 1-WK formulas for the percentage of eyes within ±0.50 D (*p* < 0.05) with SRK/T-CMAL having the highest percentage of eyes with predicted SE within ±0.50 D. The Hill-RBF formula had more accuracy in predicting eyes within ±0.50 D compared with the Holladay 1 with CMAL adjustment (*p* = 0.0143). When assessing the percentage of eyes that achieved postoperative refraction within 0.75 D of the predicted SE, the SRK/T with WK adjustment did significantly worse than the remaining formulas (*p* < 0.05) with only the exception of the Holladay 1 with WK adjustment formula, which performed as poorly as the SRK/T-WK. The BU-II, Hill-RBF, and Kane formulas could predict the postoperative manifest refraction within 1.00 D for all 35 eyes.

## 4. Discussion

In this study, we assessed the accuracy of BU-II, EVO, Hill-RBF, Holladay 1, Kane, and SRK/T in long eyes in a predominantly Caucasian population. Despite advances in formulas, surgeons are still faced with the difficulty of accurately achieving desired refractive outcomes for high myopes. In our study, the newer-generation BU-II, EVO, Hill-RBF 3.0, and Kane formulas had greater accuracy than the third-generation Holladay 1 formula with either the WK or CMAL adjustment. The SRK/T with CMAL adjustment was comparable to the newer IOL formulas. The WK axial length adjustment for SRK/T did not prove to be as accurate as the CMAL adjustment and was comparable to the accuracy of Holladay 1 (Figure 1 and Table 2).

Asians have a higher percentage of high myopia as compared with Caucasians [16,17]. As a result, most studies assessing IOL calculation accuracy within high AL have been performed in predominantly Asian study populations [14,23,24,25,26,27,28,29,30,31,32]. While our study did not have ethnicity data readily available in the paper charts, the population demographics of the study site (Draper, UT) are 91.1% white and 3.5% Asian (US Census Data for Draper, UT, available at https://www.census.gov/quickfacts/drapercityutah, accessed 27 June 2021). To the surgeon’s knowledge, only one patient was of Asian descent in the study. There have been fewer studies assessing IOL formula accuracy evaluating AL greater than 25.0 mm in predominantly Caucasian populations, and even fewer evaluating the newer vergence or artificial intelligence formulas [33,34,35]. The criteria for these studies were AL >25.0, 26.0, or 26.5 mm, whereas the current study evaluated much longer eyes. In addition to having a higher percentage of eyes with longer ALs, some studies have shown that Asians have flatter corneal shapes as compared with Caucasians [18,19]. Axial length and corneal shape are two variables known to heavily influence the predictive capabilities of IOL formulas [27]. 

Assessing the accuracy of the formulas, the MedAE of the four newer formulas, Hill-RBF, Kane, EVO, and BU-II, had the lowest prediction errors, with Kane having the lowest of all the formulas and BU-II coming in fourth. The only significant difference in MedAE identified between formulas was the Holladay 1 formula with the CMAL adjustment, which performed significantly worse than any of the four newer formulas or the SRK/T with CMAL adjustment (Table 2 and Figure 1). Similarly, in assessing the percentage of eyes that achieved a prediction error of ±0.25 D, ±0.50 D, ±0.75 D, or ±1.00 D, BU-II, Hill-RBF, EVO, Kane, and SRK/T-CMAL had the highest percentage of eyes within 0.25 D. The statistical difference showed that SRK/T-CMAL had a higher accuracy of predicting SE within 0.50 D than the Holladay 1 formulas and the WK adjustment of SRK/T. The Holladay 1 formula with WK adjustment did worse than the other remaining formulas in predicting the SE ±0.75 D (Table 2). The four new formulas also tended to predict postoperative hyperopic shift compared with the older formulas, which tended to predict a postoperative myopic shift. This result is consistent with the previous studies that have documented the tendency for newer fourth-generation formulas to lead to postoperative hyperopia as compared with older-generation formulas, which tend to have greater postoperative myopia [11].

A similar study compared the same formulas as the current study, except that they used version 2.0 of the Hill-RBF formula and looked at a population of 370 high-AL eyes of a predominantly Asian population [32]. Their study showed that the Holladay 1 with WK adjustment and Kane formulas had higher accuracy in extremely high myopes. Of note, their study defined extreme myopia as AL ≥ 30.0 mm, while the current study only had four patients that fit this criterion. However, Fuest et al. in Germany looked at eyes with long axial lengths and compared the BU-II and Hill-RBF 2.0 with Holladay 1 and SRK/T formulas and found that the BU-II and Hill-RBF 2.0 performed better than the Holladay 1 and SRK/T formulas, which was consistent with other studies, including studies consisting of Asian populations [1,34,35,36,37]. Our data support previous reports that the BU-II and Hill-RBF perform more accurately than Holladay 1 and SRK/T in long eyes. This study was able to add that the Kane, EVO, and SRK/T-CMAL formulas performed similarly to BU-II and Hill-RBF in predominately Caucasian eyes. Similar studies in four Asian populations and one European population also found newer-generation formulas such as the Kane, EVO, and BU-II to be most accurate in long eyes [38,39,40,41,42].

Our study did have some limitations: (1) Because very high axial lengths are less frequent in the population and even less so in Caucasian populations, our study sample size was not as robust as other Asian studies, and we included bilateral eyes as a result of the small size. The use of bilateral eyes can potentially compound data. (2) The chart data did not include the patient’s self-reported ethnicity, and therefore assumptions were made based on the demographics of the location of the surgical center and the physician’s recollection of presumed ethnicity. (3) The study was retrospective and we had limitations in standardization and follow-up periods. Future studies on larger sample sizes of Caucasian populations or a single-site comparison of Asian to Caucasian would be warranted.

In conclusion, our results show that for axial lengths greater than or equal to 28.0 mm, the Barrett Universal II, Emmetropia Verifying Optical, Hill-RBF version 3.0, and Kane formulas were comparable in accuracy. Additionally, the Cooke-modified axial length adjustment was better than the Wang–Koch axial length adjustment when used with the SRK/T formula. The Holladay 1 had the lowest predictive accuracy of the six formulas we tested. The most accurate prediction of high axial lengths in Caucasian eyes may be achieved with Barrett Universal II, Emmetropia Verifying Optical, Hill-RBF version 3.0, Kane, and SRK/T with the CMAL adjustment.

## Figures and Tables

**Figure 1 jcm-11-05947-f001:**
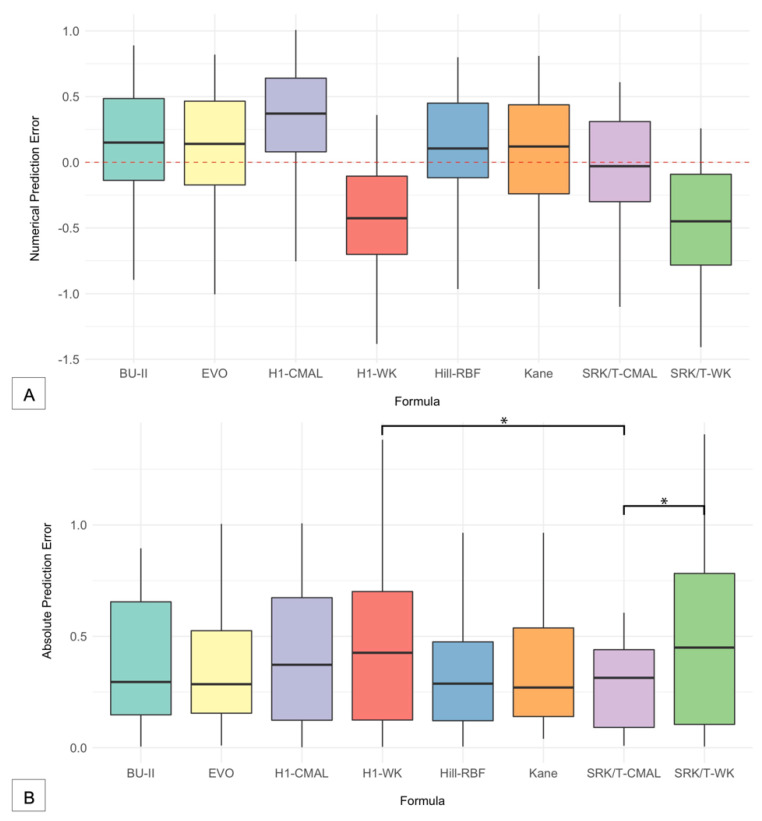
Boxplots showing the prediction errors of intraocular lens calculation formulas. (**A**) The numerical prediction errors were calculated by subtracting the predicted spherical equivalent (SE) from the postoperative SE. (**B**) The absolute prediction errors were then taken from the numerical prediction errors. BU-II = Barrett Universal II; CMAL = Cooke-modified axial length; EVO = Emmetropia Verifying Optical; H1 = Holladay 1; WK = Wang–Koch AL adjustment. * significant *p* < 0.05.

**Figure 2 jcm-11-05947-f002:**
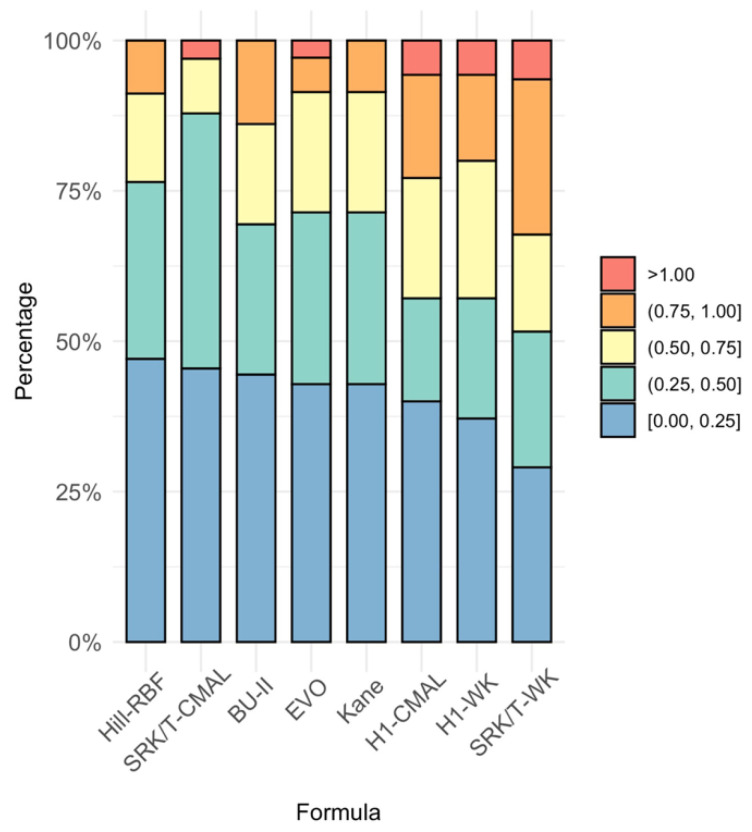
Stacked histogram comparing the percentage of eyes within ±0.25 diopters (D), ±0.50 D, ±0.75 D, and ±1.00 D of predicted spherical equivalent for various intraocular lens calculation formulas. BU-II = Barrett Universal II; CMAL = Cooke-modified axial length; EVO = Emmetropia verifying optical; H1 = Holladay 1; WK = Wang–Koch AL adjustment.

**Table 1 jcm-11-05947-t001:** Demographics, biometrics, and refractive outcomes (*n* = 35).

	*n*	(%)
Gender (F/M)	15/10	(60.0%, 40.0%)
Eye (OD/OS)	20/15	(57.1%, 42.9%)
	Mean ± SD	Range
Age, y	56.94 ± 9.56	37, 76
Axial length (mm)	28.71 ± 0.87	28.01, 31.1
ACD (mm)	3.66 ± 0.38	2.38, 4.24
Lens thickness (mm)	4.25 ± 0.52	2.96, 5.6
Average keratometry (D)	43.30 ± 1.61	41.59, 49.22
	*n*	(%)
Keratometry subgroups		
Flat (<42.0 D)	10	(28.6%)
Medium (42.0 D–46.0 D)	23	(65.7%)
Steep (>46.0 D)	2	(5.7%)
IOL Type		
Alcon MA60MA	2	(5.7%)
AMO AR40e	2	(5.7%)
enVista MX60E	1	(2.9%)
Tecnis ZCB00	24	(68.6%)
Tecnis ZCT225	2	(5.7%)
Tecnis ZXR00	4	(11.4%)
	Mean ± SD	Range
IOL power (D)	7.76 ± 3.06	–1.00, +12.00
Preoperative		
SE (D)	–11.28 ± 4.29	–18.88, −3.63
UDVA (LogMAR)	1.69 ± 0.39	0.3, 1.90
CDVA (LogMAR)	0.22 ± 0.17	0, 1.00
Postoperative		
SE (D)	–0.58 ± 0.79	–2.13, 0.75
UDVA (LogMAR)	0.20 ± 0.23	0, 0.80
CDVA (LogMAR)	0.01 ± 0.07	–0.12, 0.30
Postoperative refraction, days after surgery	147.62 ± 179.90	21, 686

D = diopters; F = female; IOL = intraocular; M = male; mm = millimeters; OD = right eye; OS = left eye; SD = standard deviation.

**Table 2 jcm-11-05947-t002:** Comparison of predictive outcomes.

Formula	MPE	SD	MAE	MedAE	Max AE	± 0.25 D ^a^	± 0.50 D ^a^	± 0.75 D ^a^	± 1.00 D ^a^
BU-II	0.146	0.451	0.379	0.295	0.895	45.71	68.57	85.71	100.00
EVO	0.147	0.416	0.361	0.285	1.005	42.86	71.43	91.43	97.14
Hill-RBF	0.136	0.407	0.333	0.288	0.965	47.06	76.47	91.18	100.00
H1-CMAL	0.352	0.393	0.419	0.370	1.010	40.00	57.14	77.14	94.29
H1-WK	−0.396	0.401	0.450	0.430	1.380	37.14	57.14	80.00	94.29
Kane	0.082	0.418	0.346	0.270	0.810	42.86	68.57	91.43	100.00
SRK/T-CMAL	−0.015	0.385	0.303	0.310	1.100	45.45	87.88	96.97	96.97
SRK/T-WK	−0.442	0.411	0.474	0.450	1.410	33.33	54.45	69.70	93.94

AE = absolute prediction error; BU-II = Barrett Universal II; CMAL = Cooke-modified axial length; D = diopters; EVO = Emmetropia Verifying Optical; H1 = Holladay 1; MAE = mean absolute prediction error; MedAE = median absolute prediction error; MPE = mean numerical prediction error; Max AE = maximum absolute prediction error SD = standard deviation; WK = Wang–Koch axial length adjustment. a = % of patients with refractive prediction errors within 0.25 D, 0.50 D, 0.75 D, or 1.00 D.

**Table 3 jcm-11-05947-t003:** Statistical analysis comparison of AE.

Formulas	BU-II	EVO	Hill-RBF	H1-CMAL	H1-WK	Kane	SRK/T-CMAL	SRK/T-WK
BU-II	-	-	-	-	-	-	-	-
EVO	0.762	-	-	-	-	-	-	-
Hill-RBF	0.189	0.442	-	-	-	-	-	-
H1-CMAL	0.09	0.114	0.097	-	-	-	-	-
H1-WK	0.408	0.207	0.156	0.801	-	-	-	-
Kane	0.158	0.172	0.974	0.073	0.164	-	-	-
SRK/T-CMAL	0.073	0.153	0.562	0.125	0.012 ^†^	0.376	-	-
SRK/T-WK	0.331	0.161	0.200	0.514	0.335	0.153	0.010 ^†^	-

AE = absolute prediction errors; BU-II = Barrett Universal II; CMAL = Cooke-modified axial length; EVO = Emmetropia Verifying Optical; H1 = Holladay 1; WK = Wang–Koch axial length adjustment. † = *p* < 0.05.

## Data Availability

The data presented are available upon request to the corresponding author. The data are not publicly available due to patient privacy.

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
