# Peer review of "Accuracy of Six Intraocular Lens Power Calculations in Eyes with Axial Lengths Greater than 28.0 mm"

_jcm, 2022, doi:10.3390/jcm11195947_

Round 1
Reviewer 1 Report
The manuscript compares intraocular lens (IOL) power calculations using six formulas for longer eyes. Having known the accurate equation for the IOL calculation will improve the surgical outcome.
The manuscript is well presented. It will add value to the manuscript if the following points are described:
1. It will be valuable to present some description about six IOL calculation formulas. For example, a table presenting all formulas with description/ differences between them.
2. What has been changed over years, particularly with newer formulas in terms of IOL calculation? Are some formulas more accurate for specific group of patients than others?
3. The manuscript title and methods present 6 formulas while results (Tables 2, 3 and Figure 2) present 8 formulas. It does not seem clear the reason behind this.
Author Response
- It will be valuable to present some description about six IOL calculation formulas. For example, a table presenting all formulas with description/ differences between them.
Thank you for your suggestion. We agree a table would be helpful, however, four of the included formulas are unpublished and are utilized via online calculators. Published formulas (Holladay 1 and SRK/T) were calculated via Excel. Because not all formulas are published, a table displaying formulas would be incomplete. Thus, we listed where calculators for formulas were available in the methods section and referenced the published formulas.
Some information describing the history of the formulas is in the introduction. Please let us know if you feel further information about each formula would be helpful- it is difficult to fully elucidate differences between each formula due to the unpublished nature of four of the six formulas.
- What has been changed over years, particularly with newer formulas in terms of IOL calculation? Are some formulas more accurate for specific group of patients than others?
Thank you for the question. This topic is discussed in the introduction and discussion sections, particularly regarding patient ethnicity and axial length (lines 65-73)_. A brief history of IOL calculations is also in the introduction (lines 44-64). We appreciate your assistance in improving the paper.
- The manuscript title and methods present 6 formulas while results (Tables 2, 3 and Figure 2) present 8 formulas. It does not seem clear the reason behind this.
Thank you for the clarification. Tables 2, 3, and Figure 2 include the SRK/T and Holladay 1 formulas in two iterations, with the Cooke-Modified axial length and the Wang-Koch axial length adjustments. Thus, there are six formulas, two of which are performed with two separate adjustments, totaling eight results. This adjustment is described in line 133. To make this clearer, we have added the following to the sentence in 134-135.
“… resulting in two iterations of each formula.”
(Please disregard uploaded file-- some line numbers are incorrect in the uploaded file and I cannot delete the file.)

Reviewer 2 Report
I read with interest the manuscript, although this study have limitations as the sample size, it is completely acceptable due to difficulty to have these eyes in the clinical practice. Authors described very well the data screening and considering the big volume of studies of formulae comparison with questionable limitations, this one has a very acceptable design. I suggest authors to conduct some changes for improving the manuscript.
P2, L48. Insert between “with high AL [11].” and “In 2019” the missed proposal before CMAL. “Fernández et al. demonstrated that the variation in prediction error with axial length was due to considering a single refractive index and not due to errors in the prediction of effective lens position (ELP), suggesting a variation in the fictitious refractive index to address this problem.”
Fernández, J., Rodríguez-Vallejo, M., Martínez, J., Tauste, A. & Piñero, D. P. New Approach for the Calculation of the Intraocular Lens Power Based on the Fictitious Corneal Refractive Index Estimation. J Ophthalmol 2019, 1–9 (2019).
Divide the Material and Methods in subsections.
2.1.- Subjects and procedures. (from P2,L72 to P2,L89), from (P3, L101 to P3, L106)
2.2.- Surgery and intraocular lenses. (from P2,L91 to P101)
3.3.- Retrospective and statistical analysis. (from P3, L107 to P147)
Bonferroni adjustement was applied for the Cochran test, but specify in the statistical section if was also applied for the Wilcoxon. Friedman test is required for multiple comparisons before the Wilcoxon for the median PE. Mark in Figure 1 through lines and asterisks over the plot which medians were statistically significant in paired comparisons since only are reported for AE in Table 3.
P2, L93 to 98. Insert the constant used for calculating the implanted IOL power following each lens name. You should include which formula/s were used to calculate the final implanted IOL power.
The 87.88% is for SRK/T-CMAL but 68.57% for Kane whereas the median AE is lower for the second one. Although this depends on distribution, just check that the calculations are correct.
P8, L288. Your study have other limitations such as: short follow-up, non-optimized constants, etc. Review “Hoffer KJ, Savini G. Update on Intraocular Lens Power Calculation Study Protocols: The Better Way to Design and Report 383 Clinical Trials. Ophthalmology”, check all your limitations that does not accomplish the described and include them in the limitations section. Furthermore, after updating the information related to the formulas used for deciding the selected IOL power, include if your study mets the invariant refraction assumption described at “Re: Hoffer et al.: Update on intraocular lens power calculation study protocols: the better way to design and report clinical trials. Fernández J, Rodríguez-Vallejo M, Piñero DP” and demonstrated at “Fernández, J., Rodríguez-Vallejo, M., Martínez, J., Burguera, N. & Piñero, D. Influence of the invariant refraction assumption in studies of formulas for monofocal and multifocal intraocular lens power calculation. Int Ophthalmol 1–8 (2022)”
Include all the required citations.
Author Response
Reviewer 2
- P2, L48. Insert between “with high AL [11].” and “In 2019” the missed proposal before CMAL. “Fernández et al. demonstrated that the variation in prediction error with axial length was due to considering a single refractive index and not due to errors in the prediction of effective lens position (ELP), suggesting a variation in the fictitious refractive index to address this problem.” Fernández, J., Rodríguez-Vallejo, M., Martínez, J., Tauste, A. & Piñero, D. P. New Approach for the Calculation of the Intraocular Lens Power Based on the Fictitious Corneal Refractive Index Estimation. J Ophthalmol 2019, 1–9 (2019).
Thank you for the suggestion and sharing this article. The suggested sentence and reference has been added (lines 48-51).
- Divide the Material and Methods in subsections.
2.1.- Subjects and procedures. (from P2,L72 to P2,L89), from (P3, L101 to P3, L106)
2.2.- Surgery and intraocular lenses. (from P2,L91 to P101)
3.3.- Retrospective and statistical analysis. (from P3, L107 to P147)
Thank you for the suggestion, the subsections have been added.
- Bonferroni adjustement was applied for the Cochran test, but specify in the statistical section if was also applied for the Wilcoxon. Friedman test is required for multiple comparisons before the Wilcoxon for the median PE. Mark in Figure 1 through lines and asterisks over the plot which medians were statistically significant in paired comparisons since only are reported for AE in Table 3.
We did not use the Bonferroni adjustment nor the Friedman test before the Wilcoxon test. The Friedman test, an extension of the Wilcoxon test, is not absolutely necessary in this particular analysis. Instead, we have presented all the p values in Table 3, of which only 2 are significant.
Thank you for the suggestion of lines and asterisk. We have added two asterisks or statistically significant comparisons to Figure 1.
- P2, L93 to 98. Insert the constant used for calculating the implanted IOL power following each lens name. You should include which formula/s were used to calculate the final implanted IOL power.
Thank you for the comments. For brevity, the IOL constants were not listed for each lens and each lens formula, but were calculated with the constants as described in the Methods section lines 135-138, listed at http://ocusoft.de/ulib/c1.htm.
- The 87.88% is for SRK/T-CMAL but 68.57% for Kane whereas the median AE is lower for the second one. Although this depends on distribution, just check that the calculations are correct.
Thank you for this observation. We have rechecked the calculations and these are correct.
- P8, L288. Your study have other limitations such as: short follow-up, non-optimized constants, etc. Review “Hoffer KJ, Savini G. Update on Intraocular Lens Power Calculation Study Protocols: The Better Way to Design and Report 383 Clinical Trials. Ophthalmology”, check all your limitations that does not accomplish the described and include them in the limitations section.
Thank you for sharing this article. We have reviewed the limitations outlined by Hoffer and feel our limitations section is in agreement.
a. For clarity regarding the ACD, as suggested by Hoffer, the following was added in line 88.“…(ACD, measured from the corneal epithelium to the lens).”
b. To further clarify the limitation of use of bilateral eyes, the following was added in line 306.
“Use of bilateral eyes can potentially compound data.” However, it is important to make full use of the limited number of eyes with high axial length in our study.
c. Optimization of the constants is not advocated by the article you suggested, in our case. Our specific study is one of the exclusions to the optimization since the optimized constants have to be from the entire population rather than that of the myopic samples.
d. We have added the non-standardization of the protocol and follow-up time, which are the usual limitations of a retrospective study. (line 310)
Furthermore, after updating the information related to the formulas used for deciding the selected IOL power, include if your study mets the invariant refraction assumption described at “Re: Hoffer et al.: Update on intraocular lens power calculation study protocols: the better way to design and report clinical trials. Fernández J, Rodríguez-Vallejo M, Piñero DP” and demonstrated at “Fernández, J., Rodríguez-Vallejo, M., Martínez, J., Burguera, N. & Piñero, D. Influence of the invariant refraction assumption in studies of formulas for monofocal and multifocal intraocular lens power calculation. Int Ophthalmol 1–8 (2022)”
Thank you for sharing these articles.
With regards to the formula used for the final IOL power of the lens inserted, this is the realm of the surgeon’s clinical judgment. The selection of the IOL formula is without preference for a single formula but an analysis of several formulas. The objective of the study is to compare the formulas but not specifically to any formula/s used for the final implanted power.
With regards the effect of the invariant refraction assumption, this may affect the analysis when we use groupings for the percentage of eyes for a particular level of PE. We addressed this by rounding -off the value of the refractions to the closest step (±0.25 D, ±0.50 D, ±0.75 D, ±1.00 D) in Table 2. We have added this description to the manuscript in lines 146-147.